# A revised model for promoter competition based on multi-way chromatin interactions at the α-globin locus

A. Marieke Oudelaar [1,2,3], Caroline L. Harrold [1,3], Lars L.P. Hanssen [1], Jelena M. Telenius [1,2], Douglas R. Higgs [1] & Jim R. Hughes [1,2]*

Specific communication between gene promoters and enhancers is critical for accurate regulation of gene expression. However, it remains unclear how specific interactions between multiple regulatory elements contained within a single chromatin domain are coordinated. Recent technological advances which can detect multi-way chromatin interactions at single alleles can provide insights into how multiple regulatory elements cooperate or compete for transcriptional activation. Here, we use such an approach to investigate how interactions of the α-globin enhancers are distributed between multiple promoters in a mouse model in which the α-globin domain is extended to include several additional genes. Our data show that gene promoters do not form mutually exclusive interactions with enhancers, but all interact simultaneously in a single complex. These findings suggest that promoters do not structurally compete for interactions with enhancers, but form a regulatory hub structure, which is consistent with recent models of transcriptional activation occurring in non-membrane bound nuclear compartments.

---

[1] MRC Molecular Haematology Unit, MRC Weatherall Institute of Molecular Medicine, Radcliffe Department of Medicine, University of Oxford, Oxford, UK. [2] MRC WIMM Centre for Computational Biology, MRC Weatherall Institute of Molecular Medicine, Radcliffe Department of Medicine, University of Oxford, Oxford, UK. [3] These authors contributed equally: A. Marieke Oudelaar, Caroline L. Harrold. *email: jim.hughes@imm.ox.ac.uk

An important question in current biology concerns the mechanisms by which genes are switched on and off during differentiation and development. Ultimately this is determined by interaction of the three fundamental regulatory elements of the genome: enhancers, promoters, and boundary elements. The activity of these elements is closely related to the three-dimensional structure of the genome. Mammalian genomes are organized in topologically associating domains (TADs), which are self-interacting regions of chromatin, usually between 100 kb and 1 Mb in size (reviewed in refs. [1,2]). The boundaries of TADs are often delineated by binding motifs for insulator proteins including CCCTC-binding factor (CTCF) and the promoters of actively transcribed genes[3,4]. There is increasing evidence that TADs are formed by a process of active extrusion of chromatin loops which is limited by these boundary elements[5–7].

Specific interactions between regulatory elements appear to occur most frequently within TADs. For example, enhancers preferentially interact with gene promoters in the same TAD[8] and disruption of TAD boundaries results in promiscuous enhancer–promoter interactions and disrupted gene activity[9–13]. However, it is not clear how specificity between multiple enhancer elements and promoters contained within a single TAD is regulated. Enhancers often exert different effects on what appear to be equally accessible genes within individual TADs. It has been proposed that enhancer-driven transcription from different promoters within a TAD is dependent on distance, orientation, or affinity of the enhancers with respect to the specific promoters[14,15]. Previous studies have suggested that enhancers may only interact with one accessible promoter at a time. This has led to a model in which the pattern of gene expression within a TAD containing multiple genes is determined by competition between promoters for limited access to shared enhancers. Based on this model, it has been proposed that co-expression of multiple genes regulated by shared enhancers in a single TAD results from rapidly alternating interactions of these genes with the enhancers in a flip–flop mechanism[16,17].

However, it has also been proposed that transcription of multiple genes might be coordinated within transcription factories[18]. Moreover, recent evidence suggests that transcriptional activation takes place in nuclear condensates, which contain a high concentration of transcription factors, co-factors, and components of the basal transcription machinery recruited by enhancer elements[19–23]. Such condensates imply that multiple regulatory elements within a TAD interact and function together in hub-like complexes. This has been postulated previously in the active chromatin hub model[24] and corresponding structures have recently been identified at a chromatin level[25,26]. In the context of these recent findings, it is unclear if and how promoter competition occurs and what the underlying structural mechanism is.

We have recently developed Tri-C, a Chromosome Conformation Capture (3C)-based approach, which can analyze multi-way chromatin interactions at single alleles[26]. Tri-C allows us to investigate whether promoters interact with enhancers in a mutually exclusive, one-to-one manner, or whether multiple promoters interact simultaneously with a shared set of enhancers in a hub structure. We have addressed this question using the well-characterized mouse α-globin locus as a model. The α-globin genes and their five enhancer elements, which fulfill the criteria for a super-enhancer[27], are located in a small TAD which is activated during erythroid differentiation[28]. We have previously shown that in vivo deletion of two CTCF-binding sites at the upstream domain boundary results in an extension of the TAD and the incorporation of three upstream genes, which become highly upregulated under the influence of the strong α-globin enhancers[13]. These mutant mice provide an excellent model to analyze the interactions between genes co-activated by a set of well-characterized enhancers in primary cells.

By performing Tri-C in erythroid cells in which the CTCF boundary is deleted, here we show that the upregulated gene promoters preferentially interact in hub-like complexes containing both the α-globin enhancers and the other active gene promoters in the domain. This shows that interactions between promoters and enhancers are not mutually exclusive and that there is no intrinsic structural competition between promoters for shared enhancers. These findings contribute to our understanding of the interplay between regulatory elements within and beyond TAD structures and the multiple layers of regulation that control gene expression.

## Results

**CTCF deletions create an extension of the α-globin domain.** We have previously defined the regulatory elements in and around the mouse α-globin cluster[13,27,28] (Fig. 1). The duplicated α-globin genes and the five globin enhancers (R1–R4 and Rm) lie within a small ~90 kb TAD. This TAD is flanked by predominantly convergent CTCF boundary elements. We have previously shown that deletion of the HS-38 and HS-39 CTCF-binding motifs causes strong upregulation of the upstream Mpg, Rhbdf1 and Snrnp25 genes in erythroid cells[13]. To investigate how this deletion influences chromatin interactions with the α-globin enhancers, we performed Capture-C from the viewpoint of the strongest enhancer element, R2, in primary erythroid cells derived from wild type (WT) mice and mice in which the CTCF-binding motifs were deleted (D3839). This shows an extension of the interaction domain in the D3839 mice, causing increased interactions between the α-globin enhancers and the Mpg, Rhbdf1, and Snrnp25 promoters (Fig. 1). The D3839 deletion thus creates an extended ~120 kb TAD in which the α-globin enhancers upregulate multiple genes. This extended domain enables us to address the mechanism by which a so-called super-enhancer interacts with multiple accessible gene promoters in a single TAD.

**The regulatory elements in the α-globin domain form a hub.** Although Capture-C produces high-resolution 3C profiles[29], it predominantly generates pair-wise interaction data. It is therefore not possible to determine the higher-order structures in which the multiple promoters and enhancers in the extended α-globin domain interact. Based on multi-way chromatin contacts generated by Tri-C, we have previously shown that the active α-globin locus is organized in a hub structure, in which multiple enhancer elements interact simultaneously with the α-globin promoters in a regulatory complex[26]. To examine this structure in the context of the extended α-globin TAD containing multiple gene promoters, we performed a Tri-C experiment from the viewpoint of the R2 enhancer in primary erythroid cells derived from D3839 and WT mice (Fig. 2 and Supplementary Fig. 1). Direct comparison of multi-way interactions between biological triplicates of D3839 and WT cells allows us to normalize and correct for 3C-related artefacts[30] and thus to robustly quantify relevant interactions. We display the multi-way interactions detected by Tri-C in contact matrices in which we exclude the viewpoint of interest and plot the frequencies with which two elements interact simultaneously with this viewpoint at a single allele. Preferential, simultaneous interactions are visible as enrichments at the intersections between these elements, whereas mutually exclusive contacts between elements appear as depletions in the matrix. Consistent with our previous findings, we observe strong, simultaneous R2 interactions with the α-globin promoter and enhancer elements in WT cells. These interactions are not decreased in the D3839 cells, as would be expected if there was competition between these

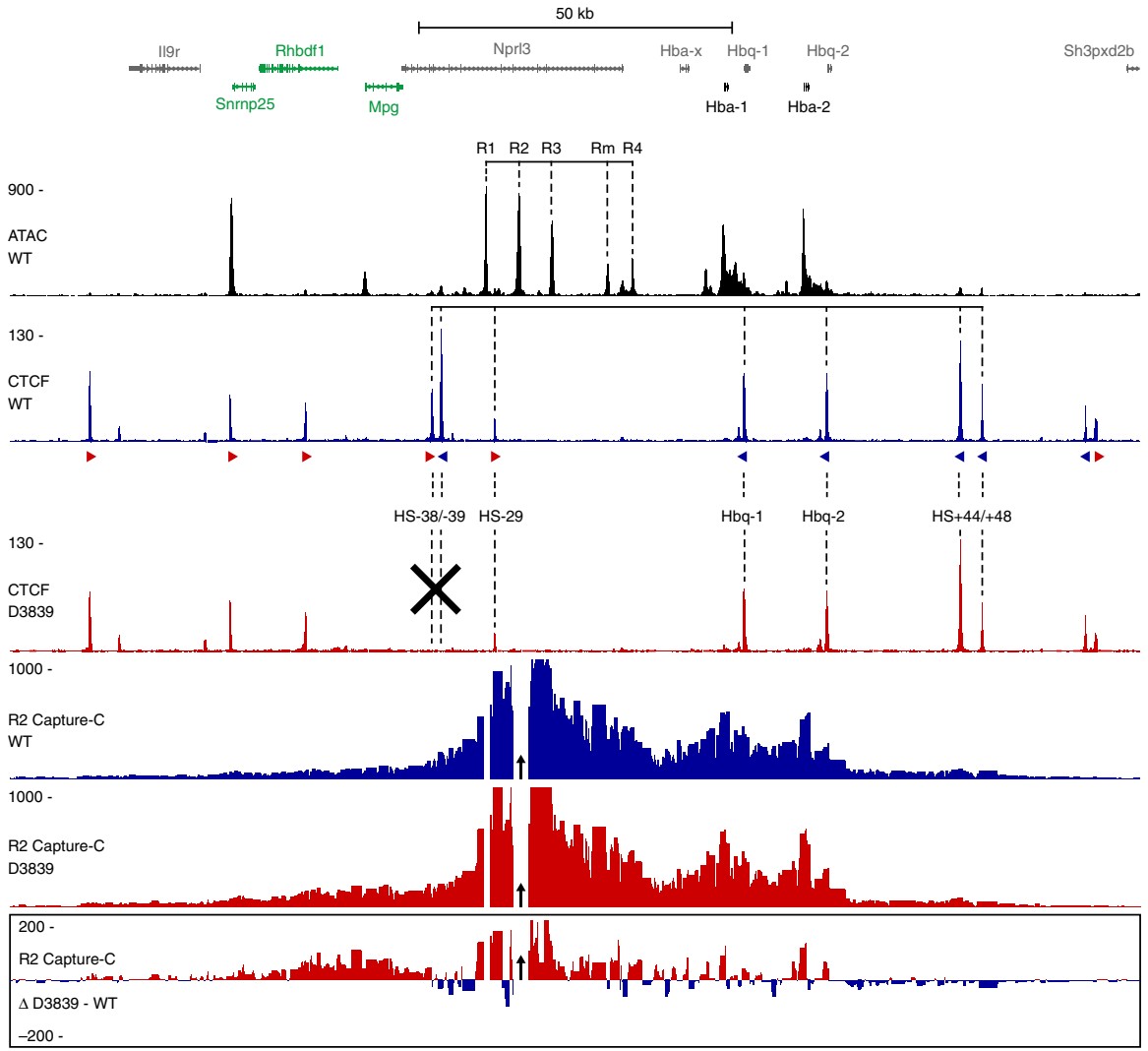

**Fig. 1** Characterization of a CTCF boundary deletion upstream of the α-globin locus. Gene annotation is shown at the top, with the α-globin genes in bold and genes upregulated by the CTCF boundary deletion highlighted in green. Open chromatin (ATAC in WT erythroid cells) is shown below, with the α-globin enhancers highlighted. CTCF occupancy in WT (blue) and D3839 (red) erythroid cells is shown underneath, with the orientation of the CTCF-binding motifs indicated by arrowheads (forward orientation in red; reverse orientation in blue). CTCF-binding sites of interest are highlighted and the deleted CTCF-binding sites are indicated with a black cross. The profiles below show Capture-C interactions from the viewpoint of the R2 enhancer (indicated with a black arrow) in WT (blue) and D3839 (red) erythroid cells, with a differential profile at the bottom. Profiles represent the mean number of normalized unique interaction counts per restriction fragment in $n = 3$ biological replicates. Coordinates (mm9): chr11:32,070,000–32,250,000.

elements. Rather, there is a trend towards increased interactions contributing to the α-globin hub in the D3839 cells, though this is not significant (Fig. 2b, c; green). In addition to the multi-way contacts between the α-globin enhancers and promoters, in D3839 we observe simultaneous interactions between the α-globin enhancer elements and the *Mpg* and *Rhbdf1* promoters (Fig. 2b, c, purple). Interestingly, the R2 contact matrix also shows simultaneous contacts with both the α-globin promoters and the *Mpg* and *Rhbdf1* promoters (Fig. 2b, gray). This indicates that all promoters in the extended D3839 TAD interact together in a single hub.

**Multiple gene promoters interact in a single regulatory hub**. To allow more extensive examination of the simultaneous interactions that occur when the upstream genes interact with the α-globin enhancers in D3839 cells, we next generated Tri-C data from the viewpoint of the *Mpg* promoter (Fig. 3 and Supplementary Figs. 2, 3). Comparison of multi-way *Mpg* interactions in

D3839 and WT cells reveals a strong increase in interactions downstream of *Mpg* after removal of the CTCF boundary. These interactions are strongest proximal to the *Mpg* promoter and reduce in strength beyond the R1 enhancer, which is located close to the HS-29 CTCF-binding site. We also observe a clear increase in more distal downstream multi-way interactions, predominantly with the regions containing the α-globin promoters and enhancers. In a dynamic flip–flop model, the *Mpg* promoter would structurally compete with the α-globin promoters for interactions with the α-globin enhancers. Such mutually exclusive interactions would be reflected by a depletion of the corresponding multi-way interactions in the Tri-C matrix. However, we find preferential interactions between these elements. For example, when *Mpg* interacts with R1, it preferentially interacts with the α-globin promoters (Fig. 3, green) and R4 enhancer (Supplementary Fig. 3). Similarly, we find enrichment of multi-way interactions between *Mpg*, R1, and the *Rhbdf1* promoter (Fig. 3, purple). This shows that *Mpg* preferentially interacts with the α-globin enhancers in a complex that contains multiple

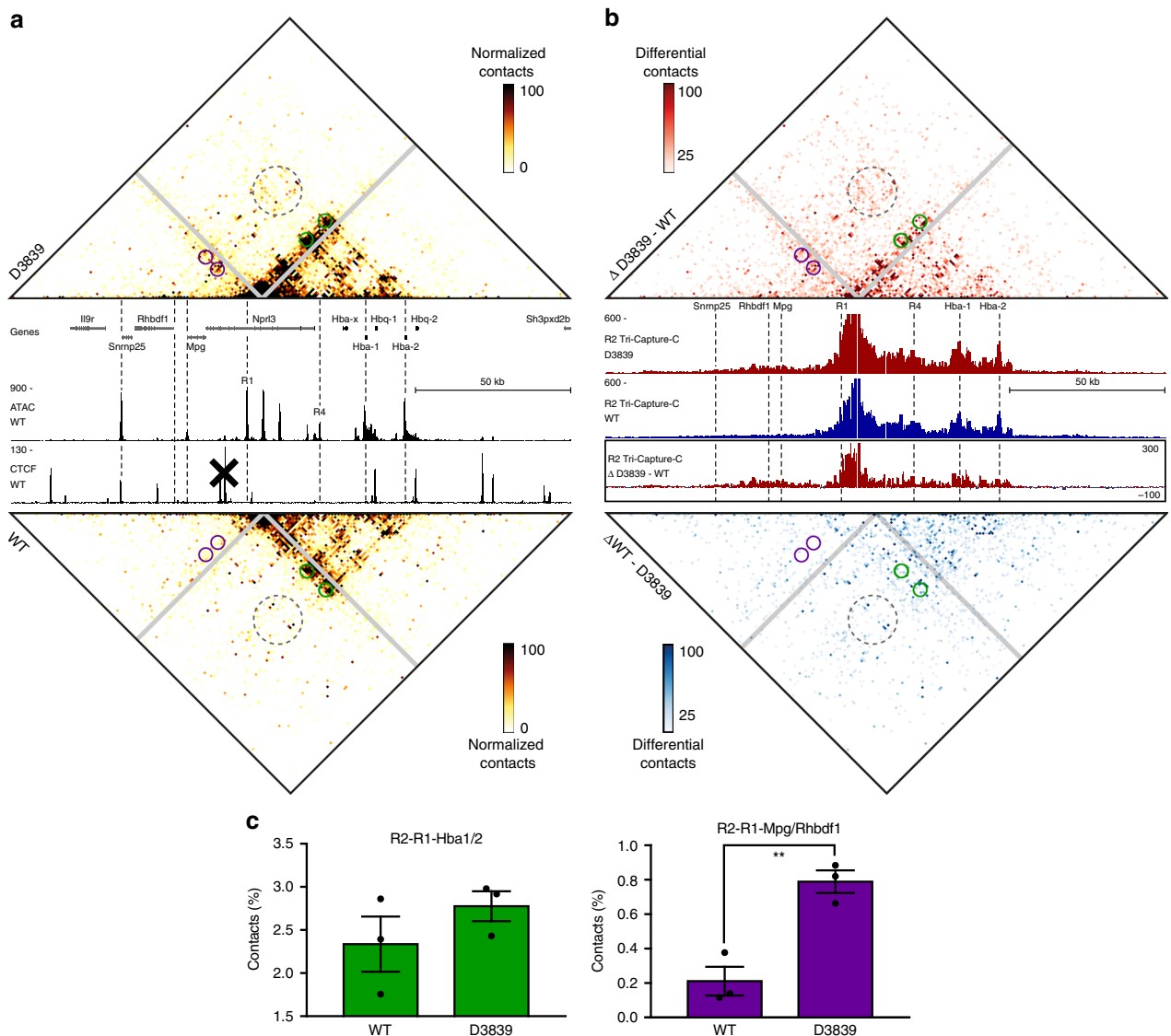

**Fig. 2** The formation of the enhancer–promoter hub at the α-globin locus is not dependent on the upstream CTCF boundary. **a** Tri-C contact matrices showing multi-way chromatin interactions with R2 in D3839 (top) and WT (bottom) erythroid cells. Matrices represent mean numbers of normalized, unique contact counts at 1 kb resolution in n = 3 biological replicates with proximity contacts around the R2 viewpoint excluded (gray diagonal). Gene annotation, open chromatin (ATAC) and CTCF occupancy in WT erythroid cells are shown in the middle. Coordinates (mm9): chr11:32,070,000–32,250,000. **b** Tri-C contact matrices showing differential multi-way chromatin interactions with R2 between D3839 and WT erythroid cells (top) and vice versa (bottom). Pair-wise interaction profiles derived from the Tri-C data from the R2 viewpoint (R2 Tri-Capture-C) are shown in the middle (D3839 in red, WT in blue), with a differential profile in the bottom panel. Coordinates (mm9): chr11:32,070,000–32,250,000. **c** Quantification of multi-way contacts between R2, R1, and the α-globin promoters (R2–R1–Hba1/2, green, P = 0.29) and R2, R1, and the Mpg and Rhbdf1 promoters (R2–R1–Mpg/Rhbdf1, purple, P = 0.0055). Quantified contacts are highlighted with corresponding colors in the matrices above. Numbers represent the proportion of these three-way contacts relative to the total in the matrix and are averages of n = 3 biological replicates, with individual data points overlaid as dot plots and the standard error of the mean denoted by the error bar. P-values were calculated by two-tailed t-tests.

enhancer elements and promoters. We also find that multi-way interactions between the three promoters are enriched (Fig. 3; orange), which further confirms that there is no structural competition between active promoters for contact with the enhancers within the extended TAD.

## Discussion

To investigate how multiple regulatory elements and genes contained within a single TAD structurally interact, we analyzed multi-way chromatin interactions in an engineered extended TAD containing the five clustered α-globin enhancers and multiple gene promoters. We show that all gene promoters interact

simultaneously with the enhancers in a common regulatory hub. Within the context of this extended TAD structure, the upstream non-globin genes do not interact as strongly and/or as frequently with the α-globin enhancers compared to the α-globin promoters. However, we show that when these genes form interactions with the α-globin enhancers, they preferentially interact in a complex in which the α-globin promoters are also present (Fig. 3, Supplementary Fig. 3). By comparing the α-globin hub in WT and D3839 cells, we show that the inclusion of additional promoters to this complex does not weaken the interactions between the α-globin promoters and enhancers and might even have an overall stabilizing effect on the hub (Fig. 2).

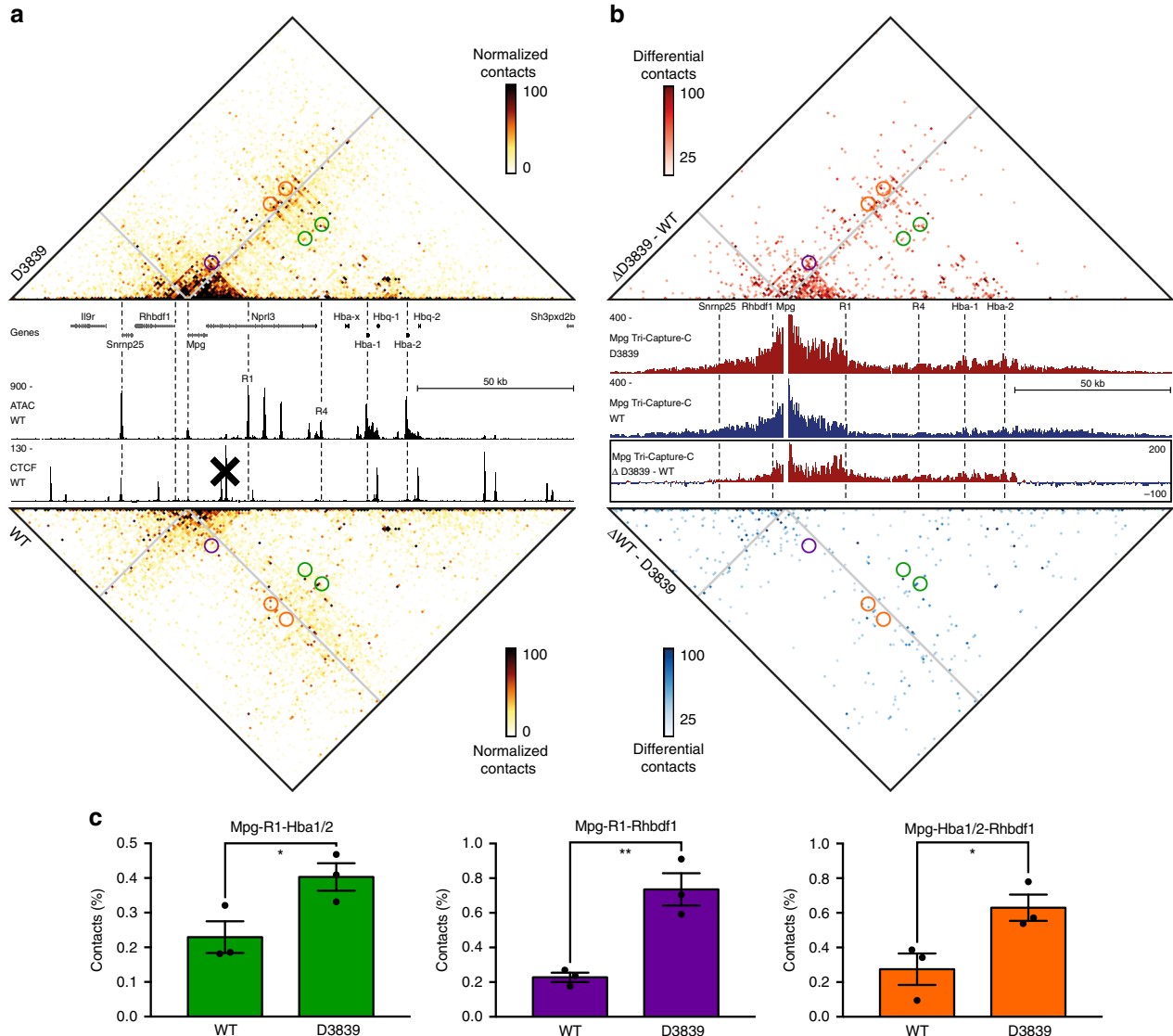

**Fig. 3** Deletion of a CTCF boundary results in the formation of a regulatory hub in which multiple gene promoters are incorporated. **a** Tri-C contact matrices showing multi-way chromatin interactions with *Mpg* in D3839 (top) and WT (bottom) erythroid cells. Matrices represent mean numbers of normalized, unique contact counts at 1 kb resolution in *n* = 3 biological replicates with proximity contacts around the *Mpg* viewpoint excluded (gray diagonal). Gene annotation, open chromatin (ATAC), and CTCF occupancy in WT erythroid cells are shown in the middle. Coordinates (mm9): chr11:32,070,000–32,250,000. **b** Tri-C contact matrices showing differential multi-way chromatin interactions with *Mpg* between D3839 and WT erythroid cells (top) and vice versa (bottom). Pair-wise interaction profiles derived from the Tri-C data from the *Mpg* viewpoint (*Mpg* Tri-Capture-C) are shown in the middle (D3839 in red, WT in blue), with a differential profile in the bottom panel. Coordinates (mm9): chr11:32,070,000–32,250,000. **c** Quantification of multi-way contacts between *Mpg*, R1 and the α-globin promoters (*Mpg*-R1-*Hba1/2*, green, *P* = 0.046); *Mpg*, R1 and the *Rhbdf1* promoter (*Mpg*-R1-*Rhbdf1*, purple, *P* = 0.0064); and *Mpg*, the α-globin promoters and the *Rhbdf1* promoter (*Mpg*-*Hba1/2*-*Rhbdf1*, orange, *P* = 0.040). Quantified contacts are highlighted with corresponding colors in the matrices above. Numbers represent the proportion of these three-way contacts relative to the total in the matrix and are averages of *n* = 3 biological replicates, with individual data points overlaid as dot plots and the standard error of the mean denoted by the error bar. *P*-values were calculated by two-tailed *t*-tests.

Our data thus show that multiple gene promoters can simultaneously interact with shared enhancers at a single allele. Our findings at the α-globin locus—a well-understood model of gene regulation—demonstrate that the previously reported flip–flop model of promoter competition, in which individual gene promoters interact with enhancers in a mutually exclusive manner, is not universally true, and that there is no intrinsic competition between gene promoters for physical access to shared enhancers within a single TAD (Fig. 4). Our model is supported by recent live-imaging experiments in Drosophila which showed coordinated bursting of two genes regulated by a single shared enhancer[31].

Our findings clarify how the activity of strong enhancers is distributed between the multiple genes surrounding these

elements. In agreement with previous findings[9–13], enhancers and promoters do not interact beyond strong CTCF-binding sites at TAD boundaries, since removal of the HS-38/-39 boundary is required for the upstream genes to be activated by the α-globin enhancers.

By contrast, within a single TAD, all promoters interact with the enhancers in a common nuclear compartment. This is consistent with previous models of transcription factories, transcriptional hubs and the recent model of transcriptional activation in nuclear condensates. However, even in the context of these cooperative structures, the activity of enhancers may not always be distributed equally between all promoters in a TAD. This might partially be explained by the relative position of

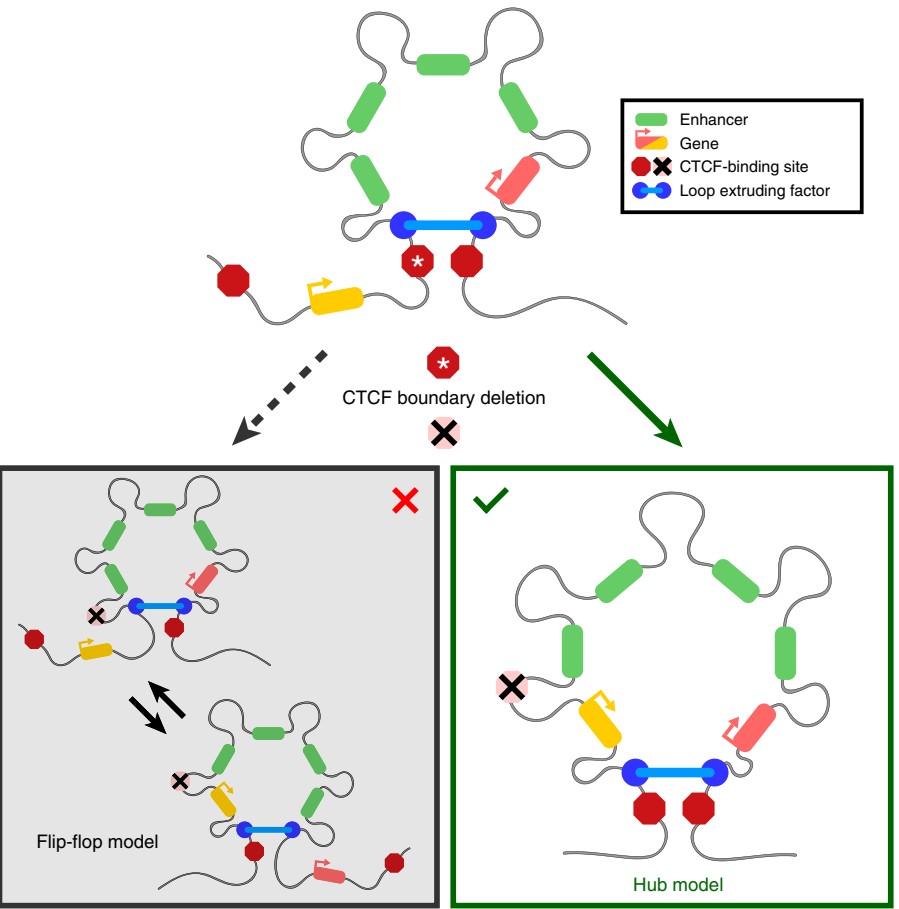

**Fig. 4** Model of the structural interplay between the regulatory elements in the α-globin locus upon removal of the upstream CTCF boundary. The CTCF boundary upstream of the α-globin enhancers (green) normally constrains their activity to the downstream α-globin genes (red). Removal of this boundary causes upregulation of the genes upstream of the α-globin enhancers (yellow). Our data show that upregulation of these genes is not caused by dynamically switching interactions between the α-globin enhancers and individual promoters (left), but by the formation of a regulatory hub in which all regulatory elements interact simultaneously (right).

promoters with respect to the enhancers. Reported examples of promoter competition have often described situations, where an active promoter located between an enhancer and another, more distal promoter causes reduced activity of the distal promoter[16,17,32,33]. It is possible that the proximal highly transcribed gene forms a barrier to loop extrusion[34], due to accumulation of large amounts of transcriptional machinery and regulatory factors. This could reduce interactions between the more distal promoter and the enhancers and hence decrease expression of the distal gene. However, the underlying mechanism is not mutual exclusivity of enhancer–promoter interactions, but a structural boundary which reduces access of the distal promoter to a cooperative hub.

Interestingly, we have shown that inclusion of the upstream genes in the α-globin hub causes upregulation of their expression, but not to the exceptionally high levels of the downstream α-globin genes[13]. This could be explained by the lower frequency of interaction and inclusion of these genes within the chromatin hub, which may correspond to a nuclear condensate. However, it is also possible that epigenetic chromatin modifications and biochemical processes within such condensates play a role, which might form another layer of regulation and potential competitive effects. For example, it could be that the α-globin genes are more responsive to the transcription and co-factors recruited by the α-globin enhancers[35].

We have previously shown that the formation of the TAD in which the α-globin enhancers and promoters interact in erythroid

cells is not dependent on the presence of all individual enhancer elements[27]. It will be of interest to further examine whether deletions of enhancer elements cause more subtle chromatin changes that might compromise the formation of the chromatin hub in both the intact locus and upon boundary deletion.

Furthermore, it will be very important to further investigate the dynamic 3D structures associated with gene activation across other gene loci and using orthogonal approaches, such as super-resolution microscopy and live-cell imaging. At the moment, imaging-based technologies are still limited in both resolution and throughput. However, with rapid technological advancements it should be possible to directly relate chromatin structures to levels of gene expression in single cells in the future. This will provide important insights into the mechanisms that control gene activity and how mutations that disrupt regulatory chromatin structures contribute to human disease.

## Methods

**Animals and cells**. We previously generated the D3839 mouse model, using TALEN and CRISPR-Cas9 to create small 19 and 26 bp deletions in the core CTCF-binding motifs of HS-38 and HS-39, respectively[13]. We performed all described experiments in primary cells obtained from spleens of female D3839 or WT C57BL/6 mice treated with phenylhydrazine, and selected erythroid cells based on the erythroid marker Ter119 using magnetic-activated cell sorting[29]. Experimental procedures were in accordance with the European Union Directive 2010/63/EU and/or the UK Animals (Scientific Procedures) Act (1986) and protocols were approved through the Oxford University Local Ethical Review process.

**Capture-C experimental procedure**. We performed Capture-C experiments in three biological replicates of primary erythroid cells derived from WT and D3839 mice following the Next-Generation Capture-C protocol[29]. We prepared 3C libraries using the DpnII-restriction enzyme for digestion. We added Illumina TruSeq adapters using NEBNext reagents and performed capture enrichment using Nimblegen reagents. We designed the capture oligonucleotides targeting the DpnII fragments containing the R1 and R2 enhancers using CapSequm[36]. Figure 1 shows the interaction profiles from the viewpoint of the R2 enhancer. Data from the R1 viewpoint have been published previously (GEO accession code GSE97871)[13]. The Capture-C libraries were sequenced on the Illumina MiSeq platform (V2 chemistry; 150 bp paired-end reads).

**Capture-C data analysis**. We analyzed Capture-C data using scripts available at https://github.com/Hughes-Genome-Group/CCseqBasicS. Because PCR duplicates are removed during data analysis, Capture-C accurately quantifies chromatin interactions[37]. The Capture-C profiles in Fig. 1 represent the mean number of unique interactions per restriction fragment from three biological replicates, normalized for a total of 100,000 interactions on the chromosome analyzed, and scaled to 1000. The differential profile highlights the interactions in D3839 cells after subtracting the normalized number of unique interactions in WT cells from those in D3839 cells. Interactions within a proximity zone of 1 kb around the viewpoint and with restriction fragments that were targeted by other capture oligonucleotides in the multiplexed capture procedure were excluded from analysis to prevent artefacts.

**Tri-C experimental procedure**. We performed Tri-C experiments in three biological replicates of primary erythroid cells derived from WT and D3839 mice following the protocol available on Protocol Exchange[38]. We used the NlaIII restriction enzyme for digestion during 3C library preparation. We added Illumina TruSeq adaptors using NEBNext DNA Library Prep reagents and Ampure XP beads (Beckman Coulter: A63881). We prepared WT libraries using the NEBNext DNA Library Prep Master Mix Set for Illumina (New England Biolabs: E6040S/L) according to the manufacturer's protocol. For each biological replicate, we performed 2–3 parallel reactions using 6 µg 3C library for sonication and all recovered material (~4.5 µg) for the subsequent library preparation. We amplified each library preparation reaction with a different index, using two separate PCR reactions per reaction (a total of 4–6 PCR reactions per biological replicate) to maximize library complexity. This procedure resulted in a total of seven technical replicates with unique indices. We prepared D3839 libraries using the NEBNext Ultra II DNA Library Prep Kit for Illumina (New England Biolabs: E7645S/L) according to the manufacturer's protocol. For each biological replicate, we sonicated 4 µg 3C library, after which we split all recovered material (~3 µg) over two parallel library preparation reactions. We amplified the two parallel reactions with the same index using two separate PCR reactions (a total of four PCR reactions per biological replicate) to maximize library complexity for each biological replicate. This resulted in a total of three replicates with unique indices. Because the Ultra II reagents are more efficient than the standard DNA Library Prep reagents, this resulted in comparable complexity for each biological replicate and similar data depth for both conditions (Supplementary Figs. 1 and 2). We pooled all libraries to enrich for viewpoints of interest in a multiplexed double capture procedure using Nimblegen reagents with custom-designed capture oligonucleotides (Supplementary Tables 1 and 2). The Tri-C libraries were sequenced on the Illumina NextSeq platform (V2 chemistry; 150 bp paired-end reads).

**Tri-C data analysis**. We analyzed Tri-C data using scripts available at https://github.com/Hughes-Genome-Group/CCseqBasicS and https://github.com/oudelaar/TriC. Briefly, we used the CCseqBasic pipeline (flags: --CCversion CS5 --nla --sonicationSize 700 --wobblyEndBinWidth 6) to perform the initial fastq processing and aligning of the data, filter out spurious ligation events and PCR duplicates, and exclude interactions with restriction fragments that were targeted by other capture oligonucleotides in the multiplexed capture procedure. We used a custom script to select reads with two or more reporters to calculate multi-way interaction counts between reporter fragments for each viewpoint. We visualized these interactions in contact matrices at 1 kb resolution, after normalizing for the total counts in each matrix and correcting for the number of restriction fragments present in each bin. We integrated this workflow in the CCseqBasic pipeline, which is available at https://github.com/Hughes-Genome-Group/CCseqBasicS. To allow for direct comparisons between WT and D3839 cells, we scaled all contact matrices to 100 normalized interactions per bin. We derived differential matrices to highlight the interactions specific for each condition after subtracting the normalized interactions in WT cells from those in D3839 cells or vice versa. We also generated regular pair-wise interaction profiles based on the total interaction counts. These Tri-Capture-C profiles were derived as described above (Capture-C—data analysis). To calculate the enrichment of multi-way interactions between regulatory elements of interest (highlighted in Figs. 2 and 3), we calculated the counts in the bins in a 2 kb radius surrounding the foci of interest in the matrix and expressed these counts as a percentage of the total number of counts in the matrix. To examine the correlation between individual replicates, we used HiCRep to calculate stratum-adjusted correlation coefficients[39], using a smoothing parameter optimized

to $h = 10$ and a maximum distance of 100,000 bp. To analyze the differences between the WT and D3839 replicates, we used unpaired, two-tailed t-tests.

**Statistical analysis**. Statistical analyses were performed with Student's two-tailed t-tests using GraphPad Prism software.

**Reporting summary**. Further information on research design is available in the Nature Research Reporting Summary linked to this article.

## Data availability
All sequencing data have been submitted to the NCBI Gene Expression Omnibus under accession number GSE130308. All other relevant data supporting the key findings of this study are available within the article and its Supplementary Information files or from the corresponding author upon reasonable request. A reporting summary for this Article is available as a Supplementary Information file.

## Code availability
Custom scripts used for the analysis of Capture-C and Tri-C data are available at https://github.com/Hughes-Genome-Group/CCseqBasicS and https://github.com/oudelaar/TriC/.

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

## Acknowledgements
We thank Matthew Gosden for technical advice and help with next-generation sequencing. This work was supported by Wellcome (Genomic Medicine and Statistics Ph.D. Program, references 105281/Z/14/Z and 109110/Z/15/Z; Chromosome and Developmental Biology Ph.D. Program, reference 099684/Z/12/Z; Wellcome Trust Strategic Award, reference 106130/Z/14/Z) and the Medical Research Council (MRC Core Funding and Project Grant, reference MR/N00969X/1). A.M.O. is funded by a Stevenson Junior Research Fellowship at University College, Oxford.

## Author contributions
A.M.O. designed experiments, performed bioinformatic analyses, and wrote the manuscript. C.L.H. performed experiments and contributed to the manuscript. L.L.P.H. performed experiments and contributed to the manuscript. J.M.T. performed bioinformatic analyses. D.R.H. co-supervised the project and wrote the manuscript. J.R.H. supervised the project and wrote the manuscript.

## Competing interests
The authors declare no competing interests.
