## [Peer Review File · Nature Communications]

Reviewers' comments:

Reviewer #1 (Remarks to the Author):

Genes can be regulated by multiple enhancers and enhancers can have multiple target genes. Heretofore, studies of communication between an enhancer and multiple genes have been limited to bivalent interactions, which does not allow to distinguish between exclusive enhancer/gene interactions with flip-flop among genes or hub-like interactions including the enhancer and more than one gene. Recent technological advances have made it possible to study multi-way contacts at single alleles. Oudelaar et al (Nature Genetics, 2018) used such an approach, Tri-C, to show that the two alpha-globin genes interact with their enhancers in a hub-like arrangement where enhancer/gene contacts are not mutually exclusive. Here, these authors study interactions within the alpha-globin cluster in mice with a deletion of CTCF sites that extend the small TAD in which the locus resides to include three neighboring genes that then fall under the regulatory influence of the alpha globin enhancer cluster. The authors succinctly show that the upregulated genes participate in hub-like complexes that include the enhancers and other active genes of the locus. The conclusion reinforces that enhancer/gene interactions are not mutually exclusive and suggest that there is no intrinsic competition between promoters for shared enhancers.

Comments:

1. Figure 2C. The value for contacts between R1/R2/Mpg-Rhbdf1 in the WT, which is across a TAD boundary, seems high. Is this a typical value for cross-TAD interaction, across the genome?
2. It is commonly thought that enhancers and target genes interact through transcription factors that contribute to the cell specificity of the interactions. Here, Mpg-Rhbdf1 come under the influence of the alpha globin enhancers in D3839 mice, presumably bypassing such requirement. Is this what the authors think? Is this what underlies the observation that Mpg-Rhbdf1 are not activated to the high levels of alpha globin transcription? What are the 'biochemical processes' referred to in the last paragraph that might be in play?

Reviewer #2 (Remarks to the Author):

This manuscript by Oudelaar and colleagues uses Tri-C (Oudelaar et al, 2018) in the a-globin locus with and without deletion of its upstream boundary (D3839 cells; Hanssen et al, 2017) to argue in favour of a model where promoters that are part of a single chromatin hub do not alternate in contacts with shared enhancers (in this case the a-globin super-enhancer locus), but rather they form a shared multi-way interaction. This has long been postulated via the concepts of the "active chromatin hub" for the global loci, and the "transcription factory" model for a number of other loci/scenarios. In fact, I think that mention to these two models should be explicitly be part of the introduction section. Nonetheless, this work provides a nice and robust documentation, the manuscript is very clearly written, and the figures nicely presented. However, the method and genetically manipulated cell system are not new, the whole postulation relies solely on experiments in one locus, that of alpha-globin, and no data from a method orthogonal to Tri-C are provided (e.g. super-resolution FISH that the Hughes lab has successfully applied in past work; Brown et al, 2018). Moreover, another critical analysis missing here is the use of lines with individual enhancers in the globing super enhancer deleted in the absence/presence of the CTCF boundary (they have also successfully managed such deletion in the past; Hay et al, 2016). This would reveal whether multiple strong regulatory elements are required to fulfil this hub-like conformation, which is what is hinted in the recent "transcriptional condensate" models. Thus, in summary, the authors touch on a question that is genuinely interesting, but only use a single locus and a single experimental setup to answer it. In my view, they would definitely need to address the issues and shortcomings mentioned above in order to make their study suitable for publication.

Additional remarks:

- The authors state that "[...] interactions are not decreased in the D3839 cells, as would be expected if there was competition between these elements. Rather, there is a trend for increased interactions contributing to the α -globin hub in the D3839 cells, though this is not significant". I am a bit skeptical as to whether 3C-based methods in general can be used in a quantitative manner. There are multiple variables along the experiment that would not allow for the direct and quantitative comparison of, in this case, Tri-C profiles. Thus, it is important to either substantiate such the quantitative nature of this and similar statements, or to tone them down altogether.

- There is one major alternative scenario that the authors have not considered here, and I think needs to be discussed. Again, as Tri-C (and 3C-based methods in general) are not quantitative, it is very difficult to rule out that what they observe as "no reduction in interactions for the alpha-globin promoter" and thus an enhancer co-sharing mode, could actually be an increased alternating looping frequency of between enhancers and promoters due to loss of insulation. In other words, it can very well be that in the absence of the boundary the globin super-enhancer makes more contacts per unit time with one or the other TSS -- and since Tri-C only scores for interaction (i.e. ligation) events (and not also for non-interaction), I cannot see how this scenario can be ruled out unless some sort of live cell imaging is used.

- Along the same lines, panes c in Figs 2 and 3 make a quantitative comparison and provide statistical confidence on the basis of a t-test. Could the authors explain why they chose to compare these contacts and for that as a fraction of the whole library (is it actually the whole sequenced library or just the contacts within the plotted matrix in cis?) and how they deal with the surrounding interaction bins in the matrix that bear little if any signal? Also, a t-test would compare the mean of the two graphs. Would the authors consider doing a Mann-Whitney test to also compare dispersion? How would that look?

- Contacts in the Tri-C triplicates presented in the Suppl. indeed look quite reproducible, but it is rather difficult to judge by eye. I suggest the authors offer some metric of the correlation between each replicate, e.g. by applying HiCRep that is used for Hi-C reproducibility on a chromosome by chromosome basis.

Reviewer #3 (Remarks to the Author):

Oudelaar et al. present evidence that the alpha-globin "super-enhancer" coordinately activates linked genes when the normal globin topological associating domain (TAD) is expanded by the deletion of a CTCF boundary element. The results are clean and clear, and the authors' interpretation of these findings in the context of a hub or condensate model is interesting. I recommend publication of the paper without further delays.

We thank the Reviewers for their helpful comments, which have guided us towards an improved presentation of our work. Please find our responses to the individual comments below.

Reviewer #1 (Remarks to the Author):

Genes can be regulated by multiple enhancers and enhancers can have multiple target genes. Heretofore, studies of communication between an enhancer and multiple genes have been limited to bivalent interactions, which does not allow to distinguish between exclusive enhancer/gene interactions with flip-flop among genes or hub-like interactions including the enhancer and more than one gene. Recent technological advances have made it possible to study multi-way contacts at single alleles. Oudelaar et al (Nature Genetics, 2018) used such an approach, Tri-C, to show that the two alpha-globin genes interact with their enhancers in a hub-like arrangement where enhancer/gene contacts are not mutually exclusive. Here, these authors study interactions within the alpha-globin cluster in mice with a deletion of CTCF sites that extend the small TAD in which the locus resides to include three neighboring genes that then fall under the regulatory influence of the alpha globin enhancer cluster.

The authors succinctly show that the upregulated genes participate in hub-like complexes that include the enhancers and other active genes of the locus. The conclusion reinforces that enhancer/gene interactions are not mutually exclusive and suggest that there is no intrinsic competition between promoters for shared enhancers.

Comments:

1. Figure 2C. The value for contacts between R1/R2/Mpg-Rhbdf1 in the WT, which is across a TAD boundary, seems high. Is this a typical value for cross-TAD interaction, across the genome?

As TADs provide about 2-fold insulation on average (see for example: Dekker & Mirny, Cell, 2016) the contact frequencies we observe for the cross-TAD R1/R2/Mpg-Rhbdf1 interactions in the WT are very consistent with the literature.

2. It is commonly thought that enhancers and target genes interact through transcription factors that contribute to the cell specificity of the interactions. Here, Mpg-Rhbdf1 come under the influence of the alpha globin enhancers in D3839 mice, presumably bypassing such requirement. Is this what the authors think? Is this what underlies the observation that Mpg-Rhbdf1 are not activated to the high levels of alpha globin transcription? What are the 'biochemical processes' referred to in the last paragraph that might be in play?

We believe that the upstream genes are recruited to the alpha-globin regulatory hub and that their expression becomes upregulated within this active environment. We don't know exactly what happens within these hubs, but we think it is possible that the alpha-globin gene promoters are more responsive to the transcription and co-factors in the hub than the upstream gene promoters. The Stark lab have recently shown that transcriptional cofactors do have specificity for different types of promoters (Haberle et al, Nature, 2019). We have expanded and clarified our speculations and included this reference in our manuscript.

Reviewer #2 (Remarks to the Author):

This manuscript by Oudelaar and colleagues uses Tri-C (Oudelaar et al, 2018) in the α -globin locus with and without deletion of its upstream boundary (D3839 cells; Hanssen et al, 2017) to argue in favour of a model where promoters that are part of a single chromatin hub do not alternate in contacts with shared enhancers (in this case the α -globin super-enhancer locus), but rather they form a shared multi-way interaction. This has long been postulated via the concepts of the "active chromatin hub" for the global loci, and the "transcription factory" model for a number of other loci/scenarios. In fact, I think that mention to these two models should be explicitly be part of the introduction section. Nonetheless, this work provides a nice and robust documentation, the manuscript is very clearly written, and the figures nicely presented. However, the method and genetically manipulated cell system are not new, the whole postulation relies solely on experiments in one locus, that of alpha-globin, and no data from a method orthogonal to Tri-C are provided (e.g. super-resolution FISH that the Hughes lab has successfully applied in past work; Brown et al, 2018). Moreover, another critical analysis missing here is the use of lines with individual enhancers in the globing super enhancer deleted in the absence/presence of the CTCF boundary (they have also successfully managed such deletion in the past; Hay et al, 2016). This would reveal whether multiple strong regulatory elements are required to fulfil this hub-like conformation, which is what is hinted in the recent "transcriptional condensate" models. Thus, in summary, the authors touch on a question that is genuinely interesting, but only use a single locus and a single experimental setup to answer it. In my view, they would definitely need to address the issues and shortcomings mentioned above in order to make their study suitable for publication.

We agree with the Reviewer that the concepts of the "active chromatin hub" and "transcription factory" should be discussed in the introduction and we have incorporated this in our manuscript. Please find our response to the additional comments in the paragraphs below:

- **One locus**

We believe that careful, well-controlled experiments in a single, well-understood model locus are extremely valuable for studying the general principles of gene regulation. In this case, the well-characterised alpha-globin locus provides an excellent system to answer our questions, as the CTCF boundary deletion provides a unique opportunity to carefully compare how multiple regulatory elements interact within an extended TAD against the background of their interactions in the WT, which controls for any other factors that could influence the 3C signal (such as proximity, mappability, etc.). It should also be mentioned that many of the current principles underpinning mammalian gene regulation were first established using the globin loci as a model. We have clarified this in our discussion.

- **No orthogonal approach**

We agree that it would be very interesting to dissect the alpha-globin hub using super-resolution FISH. Unfortunately, the alpha globin locus is very small, and it is technically extremely challenging to visualise multiple enhancer elements and genes in the alpha-globin

locus simultaneously in a multi-colour design – even using the current super-resolution FISH approach described in Brown et al. This is work in progress in the lab but unlikely to be available in the next 2-3 years. At the moment, 3C-based approaches have superior resolution compared to FISH, and we believe that our multi-way Tri-C approach offers the best resolution and sensitivity to analyse the structural conformation of all regulatory elements in the alpha-globin locus. We have clarified this in our discussion.

- **Enhancer deletions**

We agree that the question of whether multiple strong regulatory elements are required to form a chromatin hub is an interesting question to study. However, the focus of our current manuscript is how the alpha-globin super-enhancer interacts with multiple gene promoters, in its normal environment within the fully intact locus. We show that the super-enhancer interacts with multiple gene promoters in a single hub. Whether the formation of this hub is dependent on the presence of multiple enhancer elements is another interesting question, but not straightforward to answer, as all our experiments are performed in primary cells derived from mouse models. Therefore, this would require the generation of additional knock-out mice. We do not believe the generation of these mice is required to answer the question we are currently addressing in our manuscript. However, we agree with the Reviewer that it is an interesting additional question and relevant discussion point and we have incorporated it in our discussion.

In summary, we agree with the Reviewer that it would be interesting to investigate additional loci, to apply orthogonal approaches, and to examine the requirement of multiple enhancer elements for hub formation. This is work we would like to address in the future, but beyond the scope of our current manuscript. We believe that our current manuscript shows interesting data (using the most sensitive currently available method to analyse a well-characterised model locus) to address a relevant question, which are very worthwhile sharing with the community at this stage. However, we agree with the Reviewer that it is important that we do not overinterpret our data. Guided by the Reviewer's comments, we have toned down our manuscript and incorporated the many useful suggestions for follow-up work. Moreover, we have changed our title to reflect that our work is based on multi-way chromatin interaction at the alpha-globin locus.

Additional remarks:

- The authors state that "[...] interactions are not decreased in the D3839 cells, as would be expected if there was competition between these elements. Rather, there is a trend for increased interactions contributing to the α -globin hub in the D3839 cells, though this is not significant". I am a bit skeptical as to whether 3C-based methods in general can be used in a quantitative manner. There are multiple variables along the experiment that would not allow for the direct and quantitative comparison of, in this case, Tri-C profiles. Thus, it is important to either substantiate such the quantitative nature of this and similar statements, or to tone them down altogether.

3C-based methods are based on counting ligation events. We have shown in previous papers that these counts can be interpreted as a quantitative signal, as long as the method

has sufficient sensitivity and PCR duplicates are carefully removed from analysis (Davies et al, Nature Methods, 2015; Oudelaar et al, Nucleic Acids Research, 2017). Because Tri-C is based on the Capture-C method it is a very sensitive assay that generates very deep data, in which PCR duplicates are removed. Therefore, detection of ligation junctions in Tri-C is quantitative. Moreover, our experimental set-up is extremely well-controlled and designed to allow for direct comparison between the models. As described above, by comparing multi-way interactions between regulatory elements in an extended TAD with minimal changes to the wildtype (the deletions of the CTCF sites are 19 and 26 bp, see Hanssen et al), we can control for any general factors such as proximity and mappability that might affect the 3C signal. Moreover, all of our experiments are performed in triplicate biological replicates with wild-type and D3839 samples processed at the same time. To emphasise this, we have clarified our experimental set-up and the quantitative interpretation of our data in the manuscript.

- There is one major alternative scenario that the authors have not considered here, and I think needs be discussed. Again, as Tri-C (and 3C-based methods in general) are not quantitative, it is very difficult to rule out that what they observe as "no reduction in interactions for the alpha-globin promoter" and thus an enhancer co-sharing mode, could actually be an increased alternating looping frequency of between enhancers and promoters due to loss of insulation. In other words, it can very well be that in the absence of the boundary the globin super-enhancer makes more contacts per unit time with one or the other TSS -- and since Tri-C only scores for interaction (i.e. ligation) events (and not also for non-interaction), I cannot see how this scenario can be ruled out unless some sort of live cell imaging is used.

If we understand this comment correctly, the Reviewer is wondering how to interpret the finding that we do not see a reduction in interactions between the alpha-globin promoter and the super-enhancer after loss of insulation (i.e. after removal of the CTCF boundary). The Reviewer postulates that removing the boundary could cause the globin super-enhancer to make more contacts per unit time with one or the other TSS.

We are not sure if the Reviewer is suggesting that deleting the CTCF motif (45 bp of DNA in total) would change the rate with which enhancer elements interact with gene promoters and thus fundamentally change the dynamics of chromatin interactions (i.e. loop extrusion or similar mechanisms) across the entire locus. But to our knowledge there is no evidence in the literature that this might be the case. Moreover, as ligation between DNA elements in 3C requires proximity, it is hard to imagine how increasing the speed by which the enhancer interactions alternate between individual promoters would results in a 3-way ligation between these elements. To ligate together as a triplet, they need to be in physical proximity *simultaneously*.

We therefore believe that the more likely explanation for "no reduction in interactions for the alpha-globin promoter" and the detection of 3-way ligation events between these elements, is that the upstream genes join the hub in which the alpha globin promoter and enhancers interact. We provide evidence for this, as our data clearly show multi-way contacts between the alpha-globin genes, the upstream genes, and the super-enhancer. We measure these multi-way contacts at single alleles, and therefore show that multiple promoters interact with the enhancers at the same time, in the same cell.

As we discussed above, we completely agree that it is important to study the dynamics of chromatin structures using orthogonal approaches, including live-imaging, and we have included this in our discussion.

- Along the same lines, panes c in Figs 2 and 3 make a quantitative comparison and provide statistical confidence on the basis of a t-test. Could the authors explain why they chose to compare these contacts and for that as a fraction of the whole library (is it actually the whole sequenced library or just the contacts within the plotted matrix in cis?) and how they deal with the surrounding interaction bins in the matrix that bare little if any signal? Also, a t-test would compare the mean of the two graphs. Would the authors consider doing a Mann-Whitney test to also compare dispersion? How would that look?

We have measured contacts in the bins containing the relevant enhancer and promoter elements of interest. We have measured these as a percentage of all contacts in the matrix, to control for the total of interactions in the matrix. We have highlighted this in the methods. The Reviewer is right that the t-test measures the mean of the graphs. As we have 3 biological replicates, the Mann-Whitey test has insufficient power to compare dispersion between the wild-type and the D3839 model (in most software the Mann-Whitney test will always give a P value greater than 0.05 no matter how much the groups differ, if the total sample size is seven or less). However, we do show all individual data points in the graphs.

- Contacts in the Tri-C triplicates presented in the Suppl. indeed look quite reproducible, but it is rather difficult to judge by eye. I suggest the authors offer some metric of the correlation between each replicate, e.g. by applying HiCRep that is used for Hi-C reproducibility on a chromosome by chromosome basis.

We thank the Reviewer for this suggestion and have included stratum-adjusted correlation coefficients as determined by HiCRep in our manuscript.

Reviewer #3 (Remarks to the Author):

Oudelaar et al. present evidence that the alpha-globin "super-enhancer" coordinately activates linked genes when the normal globin topological associating domain (TAD) is expanded by the deletion of a CTCF boundary element. The results are clean and clear, and the authors' interpretation of these findings in the context of a hub or condensate model is interesting. I recommend publication of the paper without further delays.

Thank you!

REVIEWERS' COMMENTS:

Reviewer #1 (Remarks to the Author):

Excellent job of revision.

Reviewer #2 (Remarks to the Author):

In this revised version of the manuscript I previously saw, Oudelaar et al. did not really add (1) data from another locus or (2) an orthogonal approach to validate their findings. I am in agreement with their argument that the alpha-globin locus is probably too small for microscopy; I am also convinced by most of their explanations/clarifications, as well as with the way the text has been restructured. I find that the Hughes lab consistently does a careful job in implementing and analysing Capture-C, and I am of the view that the data shown here are indeed high quality and carefully interpreted.

As a result, I endorse publication of the manuscript, pending the examination of multi-way contacts in a different locus (which I still think would vastly improve the paper), the decision for which I leave at the editors' discretion.

REVIEWERS' COMMENTS:

Reviewer #1 (Remarks to the Author):

Excellent job of revision.

Thank you.

Reviewer #2 (Remarks to the Author):

In this revised version of the manuscript I previously saw, Oudelaar et al. did not really add (1) data from another locus or (2) an orthogonal approach to validate their findings. I am in agreement with their argument that the alpha-globin locus is probably too small for microscopy; I am also convinced by most of their explanations/clarifications, as well as with the way the text has been restructured. I find that the Hughes lab consistently does a careful job in implementing and analysing Capture-C, and I am of the view that the data shown here are indeed high quality and carefully interpreted.

As a result, I endorse publication of the manuscript, pending the examination of multi-way contacts in a different locus (which I still think would vastly improve the paper), the decision for which I leave at the editors' discretion.

Thank you. We agree that it would be interesting to examine multi-way contacts in other loci. However, it is important to keep in mind that the mouse α -globin locus is extremely well-characterised and that the availability of a mouse model with a CTCF deletion offers a unique experimental set-up. Therefore, providing matching data on additional loci would require us to generate new mouse models and thus cause a substantial delay. As the mouse α -globin locus has been used as a model for general principles of gene regulation for decades, we believe the study of additional loci is not necessary and beyond the scope of our manuscript.